# N-Terminal Pro-B-Type Natriuretic Peptide (NT-proBNP)—A Prognostic Biomarker in Older and/or Frail Adults with Advanced Gastroesophageal Cancer: A Post Hoc Analysis of the GO2 Clinical Trial

**DOI:** 10.3390/cancers17040601

**Published:** 2025-02-10

**Authors:** Yuewei Tao, Chim C. Lang, Russell D. Petty, Peter S. Hall, Mark A. Baxter

**Affiliations:** 1Dundee Medical School, Ninewells School of Medicine, University of Dundee, Dundee DD1 9SY, UK; 2406970@dundee.ac.uk; 2Department of Cardiology, Ninewells Hospital, Dundee DD1 9SY, UK; c.c.lang@dundee.ac.uk; 3Division of Cardiovascular Research, Ninewells School of Medicine, University of Dundee, Dundee DD1 9SY, UK; 4Division of Cancer Research, Ninewells School of Medicine, University of Dundee, Dundee DD1 9SY, UK; r.petty@dundee.ac.uk; 5Tayside Cancer Centre, Ninewells Hospital, Dundee DD1 9SY, UK; 6Edinburgh Cancer Research Centre, University of Edinburgh, Edinburgh EH8 9YL, UK; p.s.hall@ed.ac.uk

**Keywords:** gastroesophageal cancer, frailty, geriatric oncology, sarcopenia, biomarkers

## Abstract

Aging is associated with increasing comorbidity and frailty, all of which are associated with poorer cancer outcomes. Older adults with cancer are often excluded from clinical trials; therefore, we lack data to inform our decision-making appropriately. This is particularly relevant in poor prognosis cancers such as those of the upper intestinal tract. The identification of novel prognostic biomarkers may help personalise management plans. One such potential biomarker is serum N-terminal pro-B-type Natriuretic Peptide (NT-proBNP). Serum NT-proBNP can be used as a measure of cardiac health and function and has established links with sarcopenia, which is prognostic in cancer. We therefore aimed to investigate the role of NT-proBNP as a prognostic biomarker in advanced gastroesophageal cancer using the data from a completed clinical trial in older and/or frail patients (the GO2 trial). The GO2 trial investigated the role of chemotherapy dose de-escalation in older and/or frail patients. Our results demonstrate an association between pre-treatment levels of NT-proBNP (cBNP) and overall survival; those with higher levels had poorer survival. We also found that there was no clear association between age, patient fitness and NT-proBNP. These findings suggest that NTproBNP may warrant further investigation as a prognostic biomarker and as an adjunct to decision-making in clinics.

## 1. Introduction

Gastroesophageal cancer is a leading cause of cancer-related mortality worldwide, causing approximately 1.3 million deaths annually [1]. The prognosis for patients diagnosed with biomarker-negative advanced gastroesophageal cancer remains poor, with a median survival of less than one year [2]. Despite the recent emergence of biomarker-directed targeted therapies, cytotoxic chemotherapy continues to be a cornerstone of treatment for advanced gastroesophageal (aGO) cancer [3].

The majority of patients with aGO cancer in the United Kingdom (UK) are aged over 70 at diagnosis [4]. Increasing age is associated with the accumulation of comorbidity, deficits in geriatric domains and, ultimately, frailty [5,6]. These factors complicate treatment decisions and can contribute to poorer cancer outcomes [7].

This has added importance for older adults as several studies have demonstrated that the priorities of older adults with cancer can differ from the younger, fitter patients normally recruited to clinical trials. These priorities include prioritising function, independence and cognition, often over survival [8,9]. Therefore, it is crucial to identify potential prognostic biomarkers that can help guide discussion, shared decision-making and therapeutic strategies in an older population.

One of the symptoms of aGO cancer is reduced oral intake, which, alongside reduced activity, can result in sarcopenia. Sarcopenia, along with malnutrition and cancer cachexia, has been identified as a significant prognostic factor in cancer, including in patients with gastroesophageal cancer [10,11]. There are established links between peripheral muscle health (captured by sarcopenia) and cardiac muscle health and function—termed the “muscle hypothesis” [12].

A biomarker of cardiac muscle health, used widely in the diagnosis and management of heart failure, is N-terminal pro-B-type Natriuretic Peptide (NT-proBNP). NT-proBNP is a bioproduct of the plasma brain natriuretic peptide (proBNP) breakdown, which is cleaved into the biologically active BNP (which is released in response to a volume overload and increased ventricular wall tension) and the biologically inactive NT-proBNP. An increase in NT-proBNP levels is commonly associated with myocardial stress, which further correlates with adverse outcomes in cardiovascular diseases [13].

NT-proBNP has been shown to be elevated in sarcopenia in health [14] and in patients with heart failure with reports of an inverse relationship between axial muscle mass and NT-proBNP [15]. The mechanisms underpinning how muscle mass may relate to natriuretic peptides are unclear. Both NT-proBNP and sarcopenia are prognostic in heart failure and other diseases [16,17]. While NT-proBNP is elevated in cancer [18], even in the absence of cardiac disease, the association with cancer prognosis has not been well investigated.

The GO2 clinical trial enrolled older and/or frail patients with advanced gastroesophageal cancer to investigate the role of chemotherapy dose de-escalation [19]. Dose-reduced doublet chemotherapy (60% of full-dose capecitabine/oxaliplatin) was found to have non-inferior progression-free and overall survival compared to full the dose, with an improved overall treatment utility (a composite outcome measure designed to quantify the effect of palliative treatments on individuals).

The population recruited to the GO2 trial was felt to better represent real-world experience than traditional trial populations. Within the trial, patients underwent baseline assessment, including detailed frailty assessments and NT-proBNP measurement, providing an opportunity to explore the prognostic role of NT-proBNP and its relationship with frailty.

In this post hoc analysis, we aimed to investigate the prognostic significance of baseline NT-proBNP levels in patients involved in the GO2 trial as well as investigate any association between NT-proBNP and frailty. We hypothesised that an elevated baseline NT-proBNP would be associated with inferior overall survival in those who received systemic chemotherapy.

## 2. Materials and Methods

### 2.1. Patient Population

The GO2 trial (ISRCTN44687907) recruited 559 patients, each undergoing comprehensive baseline assessments. These included a blood assessment (including optional NT-proBNP), as well as a clinical assessment that covered the detailed assessment of geriatric domains (including calculation of the GO2 frailty score, which was a categorical classification, grouping patients into non-frail (deficits in zero to one domain), mildly frail (deficits in two domains) and severely frail (deficits in three or more domains)).

Inclusion criteria for this post hoc analysis were (1) availability of baseline NT-proBNP measurements, (2) receipt of at least one cycle of chemotherapy as part of the GO2 trial and (3) complete baseline data for Eastern Cooperative Oncology Group Performance Status (ECOG PS) and/or the GO2 frailty grouping.

### 2.2. NT-proBNP Measurement

Baseline NT-proBNP levels were measured before starting chemotherapy. Blood samples were analysed and NT-proBNP levels were reported in picograms per millilitre (pg/mL) at local laboratories’ reference limits. To standardise the results for our analysis, a corrected NT-proBNP (cBNP) was calculated by dividing the NT-proBNP value by the upper limit of normal in each centre. Values of cBNP > 1.0 were considered abnormal. For the purpose of analysis, patients were divided into normal and above-normal (or abnormal) cBNP as well as into low (bottom 25%), middle (25–75%) and high (top 25%) cBNP levels.

### 2.3. Statistical Analysis

The main endpoint of this analysis focused on overall survival (OS), defined as the time from randomisation in the GO2 trial to death by any cause. Patients were investigated either at 12 months or at their last known follow-up date. Kaplan–Meier survival curves were generated for each cBNP group, and differences in survival were evaluated using the log-rank test.

Cox proportional hazard regression models were used to estimate hazard ratios (HRs) for death, with 95% confidence intervals (CI). The models were adapted for GO2 trial stratification factors, including age, sex, ECOG PS, chemotherapy dose, presence of metastases and cBNP group. Sensitivity analyses were performed to explore the interaction between cBNP levels and frailty to elucidate the value of cBNP across different levels of frailty.

Two-sided statistical tests with a significance *p*-value < 0.05 were carried out in our study. R version 4.0.2 (R Foundation for Statistical Computing, Vienna, Austria) was used for data analysis.

## 3. Results

### 3.1. Patient Characteristics

A total of 241 patients who received chemotherapy had a baseline NT-proBNP in the GO2 trial and were included in this post hoc analysis. Demographics of the included cohort are shown on Table 1. The median age of the cohort was 76 years old (range 52–89 years old), with 77.6% (187 patients) male and 87.6% with adenocarcinoma. The majority of the patients were ECOG PS 1 (53.1%) and had deficits in at least two geriatric domains (79.7%). This population was representative of the overall recruited GO2 patient population (Appendix A). Comorbidity data were unavailable but within the included population, 123 (51%) were unable to complete the timed up-and-go test within 10 s, 172 (71.4%) were taking five or more medications and 11 (4.6%) reported two or more falls in the previous 6 months.

### 3.2. Overall Survival

Within the included cohort, the median overall survival (mOS) was 7.1 months (95% CI: 6.2–7.9), with mOS 7.9 months (95% CI: 6.7–10.4), 5.4 months (95% CI: 4.5–8.2) and 7.1 months (95% CI 5.8–8.1) in dose levels A (100% OX), B (80% OX) and C (60% OX), respectively. Dose-reduced chemotherapy (60% OX, which is now a standard of care) was non-inferior to the full dose (HR 1.21, 95% CI 0.84–1.74, *p* = 0.3). These findings are in line with those reported in the GO2 trial.

Improved ECOG PS was associated with a numerically better prognosis, but this did not reach significance; mOS was 7.9 vs. 7.5 vs. 5.8 months in ECOG PS of 0, 1 and 2 or more, respectively. In contrast, a poorer GO2 frailty score was associated with inferior survival; mOS was 9.4 vs. 8.2 vs. 5.8 months in the non-frail, slightly frail and severely frail groups. The HR for the severely frail group was 2.05 (95% CI; 1.38–3.04, *p* < 0.001).

### 3.3. NT-proBNP, Patient Survival and Frailty

Among the 241 patients analysed, 80 (33.2%) had NT-proBNP levels above the local ULN (Table 1 and Appendix A). Those with an elevated cBNP at baseline had a poorer mOS (Figure 1); 5.2 months vs. 7.9 months (HR 1.57, 95% CI: 1.16–2.13, *p* = 0.004). Further survival analysis to determine the impact of the level of elevation demonstrated a significant difference in the overall survival (OS) according to cBNP level (i.e., low, medium, high). Patients in the highest cBNP group had the poorest survival outcomes, with a median OS of 5.3 months compared to 6.8 months in the medium cBNP group and 8.2 months in the low cBNP group (HR = 1.57, 95% CI: 1.04–2.37, *p* = 0.031) (Figure 2). This association remained significant in a multivariate Cox regression analysis adjusting for trial stratification factors and albumin (HR = 1.69, 95% CI: 1.11–2.57, *p* = 0.01) compared to the low-cBNP group, as shown in Figure 3.

There was no significant observed difference in corrected NT-proBNP (cBNP) levels according to either age group (<75 vs >75 years, *p* = 0.885) or ECOG PS (*p* = 0.522) (Table 2). In addition, while cBNP increased with increasing frailty as measured by the GO2 frailty grouping, this did not meet significance (*p* = 0.706), suggesting that there was no clear relationship (Table 2). An analysis of the survival impact of chemotherapy dose level according to baseline BNP suggested that there was no significant impact of dose in either those with a normal or elevated cBNP.

## 4. Discussion

The cancer population is ageing worldwide [20]. The older population encountered in clinical practice differs from those recruited to traditional clinical trials—often with greater comorbidity and frailty [21]. It is therefore necessary to extrapolate data from these trials, with the risk of either over or undertreating older adults with cancer [22]. This can result in poorer cancer outcomes and, in particular, increased toxicity, impaired quality of life and poorer survival. There is an urgent need to identify prognostic biomarkers, which could improve shared decision-making and enable a more personalised management plan. This is particularly relevant for patients with aGO cancer due to their often high symptom burden at diagnosis, the toxicity associated with treatment and inherently poor prognosis [19,23].

Given the link between peripheral muscle health (a marker of sarcopenia, which is prognostic in cancer) and cardiac health, we sought to investigate the potential prognostic role of baseline levels of the cardiac biomarker NT-proBNP in older and/or frail patients with advanced GO cancer. The included population, from a completed clinical trial, had a high level of baseline frailty (majority ECOG PS 1 and deficits in at least two geriatric domains) and a median age of 76—representing real-world experience [24,25].

In our study, we found that despite not having documented significant cardiovascular disease, a third of patients had an elevated NT-proBNP at baseline. We then investigated the impact of an elevated baseline NT-proBNP on patient survival and demonstrated significant differences in overall survival according to cBNP. Those individuals with the highest cBNP (defined as the top 25% for the purpose of analysis) had the poorest survival outcomes, with a difference of almost 3 months compared to those with the lowest cBNP. This survival pattern remained significant after key confounding factor adjustments (HR = 1.67, *p* = 0.016).

Our findings suggest that both an elevated cBNP and the level of elevation and associated with inferior survival in patients with advanced GO cancer treated with chemotherapy. This observation is similar to what has been previously reported in other medical conditions [26,27] and in the supportive care setting in oncology [28]. These results suggest that NT-proBNP warrants further investigation as a potential prognostic biomarker.

One possible reason for this observation may be sarcopenia, which is prognostic in cancer and associated with frailty. Given the established links with sarcopenia, an increased NT-proBNP level may act not only as a biomarker of cardiovascular muscle health but may also reflect broader aspects of frailty and physiological decline in older adults with cancer [29]. Such findings highlight the clinical potential of NT-proBNP in stratifying older adults based on their oncological and cardiovascular risks.

Despite this hypothesis, no clear associations were found between corrected NT-proBNP (cBNP) levels and the frailty measures used in the GO2 trial. An explanation for the lack of association may be the heterogeneity that exists within the patient population and a lack of granularity within the categorical groupings of ECOG PS and the GO2 frailty groups.

As such, serum NT-proBNP may provide unique prognostic information in older and/or frail patients that might not be provided via traditional clinical assessment and frailty tests. In clinical practice, fitness is often determined using ECOG performance status or the Rockwood Clinical Frailty Scale. However, these may not fully capture the complexity of ageing and comorbidity when deciding to deliver treatments [30].

As a biomarker that shows cardiovascular health, NT-proBNP could introduce another layer to frailty assessment, aiding in a more individualised patient management approach in clinics. For instance, a patient with high NT-proBNP levels might be better with a less aggressive chemotherapeutic treatment plan (including supportive care alone).

The findings of this study, while needing validation, raise questions regarding the underlying mechanisms linking NT-proBNP to survival. It is possible that increased NT-proBNP levels might indicate a combination of factors, such as sarcopenia, cachexia and chronic inflammation. These are commonly known to influence cancer prognosis in older adults. Further research is therefore needed to reveal such mechanisms and whether using NT-proBNP screening could identify patients who might benefit from more tailored therapeutic managements, i.e., targeted prehabilitation, cardioprotective interventions or dose-modified chemotherapy regimens.

The main strength of this study is that it is a post hoc analysis from a completed clinical trial, assuring that the baseline and follow-up data are reliable. The NT-proBNP data and frailty assessments in this large cohort allow us to perform an in-depth exploration of the prognostic value of NT-proBNP. In addition, our patient cohort is highly representative of the older and/or frail patients with aGO cancer in real-world clinical practice, making our findings clinically relevant. This study is also one of the first to evaluate NT-proBNP as a prognostic biomarker in this cancer population, highlighting its novelty in the field.

Despite the above strengths, this study has its limitations. We only measured NT-proBNP levels at baseline, and changes in NT-proBNP over the course of chemotherapeutic treatment were not assessed. Longitudinal monitoring might provide more information regarding NT-proBNP acting as a biomarker for future disease progression assessments. We must acknowledge that the detailed exploration of the relationship between frailty, sarcopenia, NT-proBNP and survival cannot be carried out due to the lack of formal assessment of sarcopenia. In addition, while the study population was representative of an older and frail cancer population, the findings may not be applicable to other patient populations, such as younger and fitter patients with aGO cancer. Therefore, external studies will be required in the future.

## 5. Conclusions

In summary, in our study, NT-proBNP was found to be an independent prognostic marker in older and/or frail patients with aGO cancer. Using it in clinical practice might provide a way to improve patient outcomes through more accurate patient assessment prior to management decisions. Future research is needed to confirm these results and to further explore NT-proBNP in cancer management. If a role for NT-proBNP is confirmed, this may provide a rationale for studies investigating the role of cardioprotective medications such as beta-blockers and ACE inhibitors alongside standard of care.

## Figures and Tables

**Figure 1 cancers-17-00601-f001:**
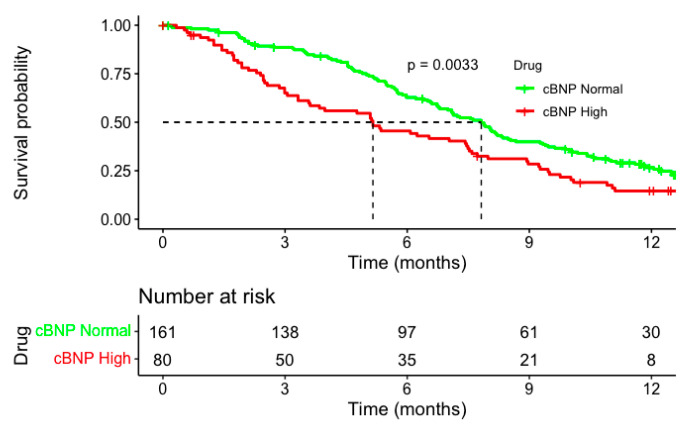
The Kaplan–Meier survival curves demonstrating the OS stratified by corrected NT-proBNP (cBNP) levels among advanced gastroesophageal cancer patients. Patients were divided into two groups: normal cBNP (green line) and high cBNP (red line).

**Figure 2 cancers-17-00601-f002:**
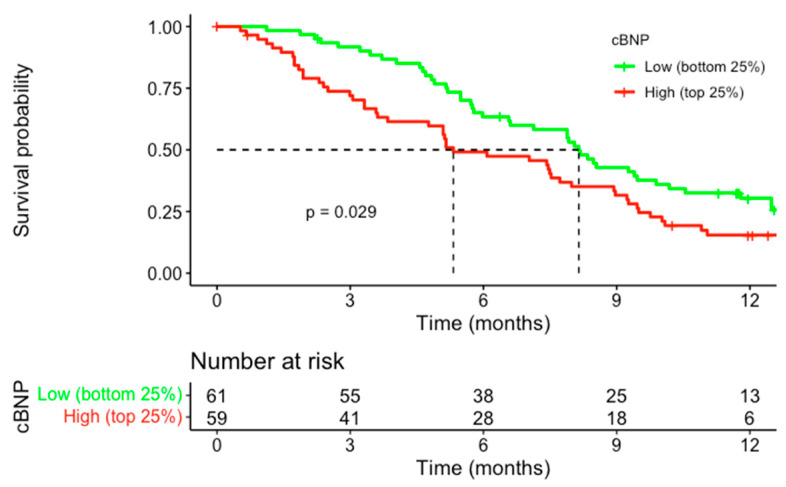
The Kaplan–Meier survival curves demonstrated the OS stratified by corrected NT-proBNP (cBNP) levels among advanced gastroesophageal cancer patients. Patients were divided into two groups: low cBNP (bottom 25%, green line) and high cBNP (top 25%, red line).

**Figure 3 cancers-17-00601-f003:**
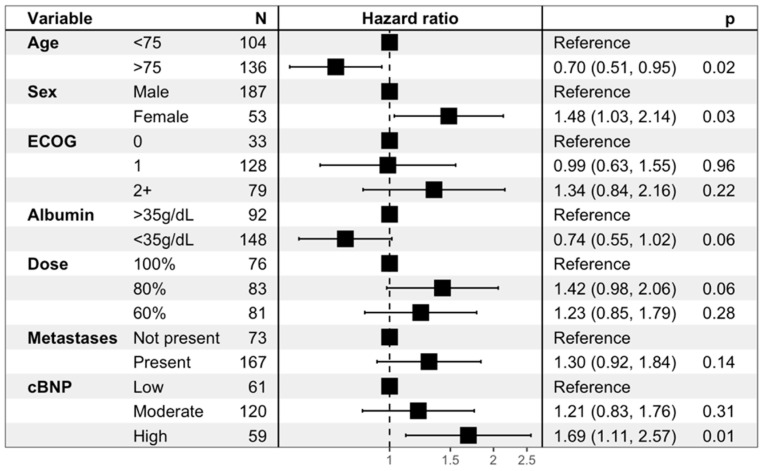
Forest plot of hazard ratios (HR) for OS by baseline characteristics and cBNP levels. The forest plot presents the hazard ratios (HRs) for overall survival, along with 95% confidence intervals (CIs).

**Table 1 cancers-17-00601-t001:** Baseline demographics and clinical characteristics of patients (*n* = 241).

Characteristic	Value
Age, years (median, range)	76 (52–89)
Sex, *n* (%)	
Male	187 (77.6%)
Female	54 (22.4%)
Histology, *n* (%)	
Adenocarcinoma	211 (87.6%)
Squamous	30 (12.4%)
Metastases	
Not present	73 (30.3%)
Present	168 (69.7%)
ECOG Performance Status, *n* (%)	
0	33 (13.7%)
1	128 (53.1%)
2+	80 (33.2%)
Albumin	
>35 g/dL	148 (61.4%)
<35 g/dL	92 (38.2%)
Haemoglobin (g/dL)	12.1 (6.6–18.0)
GO2 Frailty Grouping	
No frailty (Deficit in zero to one domain)	49 (20.3%)
Mild frailty (Deficit in two domains)	57 (23.7%)
Severe frailty (Deficit in three or more domains)	135 (56.0%)
Corrected NT-proBNP Levels, *n* (%)	
Elevated	80 (33.2%)
Normal	161 (66.8%)

**Table 2 cancers-17-00601-t002:** Corrected BNP (cBNP) expressed as median (range) according to age group, ECOG performance status and GO2 frailty group.

	Group	Baseline cBNP	*p*-Value
		(Median, Range)	
Age Group	<75 (*n* = 104)	0.65 (0.03, 10.3)	0.885
	≥75 (*n* = 136)	0.70 (<0.005, 9.29)	
ECOG PS	0 (*n* = 33)	0.84 (0.09, 6.89)	0.522
	1 (*n* = 128)	0.59 (<0.005, 7.24)	
	2+ (*n* = 79)	0.71 (0.03, 10.3)	
GO2 Frailty Group	No frailty (*n* = 49)	0.53 (0.09, 7.70)	0.706
	Mild frailty (*n* = 58)	0.65 (<0.005, 9.29)	
	Severe frailty (*n* = 134)	0.73 (0.033, 10.3)	

## Data Availability

Data are available on request to the corresponding author.

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
