# Peer review of "N-Terminal Pro-B-Type Natriuretic Peptide (NT-proBNP)—A Prognostic Biomarker in Older and/or Frail Adults with Advanced Gastroesophageal Cancer: A Post Hoc Analysis of the GO2 Clinical Trial"

_cancers, 2025, doi:10.3390/cancers17040601_

Round 1
Reviewer 1 Report
Comments and Suggestions for Authors
The authors provide an exploratory analysis of a previously published GO2 trial (originally published in 2021) aimed at a smaller cohort with available cBNP levels. The study is original, the topic is important, and the conclusions are supported by data and relevant.
I do have, however, many important questions. I also wish the authors to provide better baseline tables.
COMMENTS:
1. Please provide an appendix comparison of the selected patient to the entire GO2 trial. And unselected patients (241 selected, 5959 the original cohort).
2. Please provide actual proBNP levels in the appendix, including an analysis of ULN for each participating center. Otherwise, the reader may not be able to practically compare to their own institution
3. Other predictors of survival are known, including as simple as serum albumin level, and hemoglobin. These need to be published.
4. Assessment of frailty and sarcopenia often includes grip tests or other assessments. Were any of these used beyond GO2 Frailty Grouping?
5. Please list patients’ comorbidities, including cardiac.
6. Figure 3: Please explain how older patients have IMPROVED overall survival. I would expect worsened survival.
7. Figure 3: The female sex is usually protective, but here it is worsening survival. Please recheck the data.
8. How do you explain that metastatic vs non-metastatic cancer did not experience any difference in survival?
9. What were the causes of mortality: cancer vs. comorbidity?
10. Please shorten discussion
Author Response
Reviewer 1:
The authors provide an exploratory analysis of a previously published GO2 trial (originally published in 2021) aimed at a smaller cohort with available cBNP levels. The study is original, the topic is important, and the conclusions are supported by data and relevant.
I do have, however, many important questions. I also wish the authors to provide better baseline tables.
COMMENTS:
Dear Reviewer 1 – thank you for taking the time to review our manuscript. We have taken your comments onboard and we feel they have significantly improved the manuscript. Please find below our replies.
- Please provide an appendix comparison of the selected patient to the entire GO2 trial. And unselected patients (241 selected, 5959 the original cohort).
Thank you – this has now been added comparing to the 558 patients in the whole trial.
- Please provide actual proBNP levels in the appendix, including an analysis of ULN for each participating center. Otherwise, the reader may not be able to practically compare to their own institution
Thank you. We have added this including the actual value, lower limit of normal and upper limit of normal.
- Other predictors of survival are known, including as simple as serum albumin level, and hemoglobin. These need to be published.
Thank you. We have incorporated Albumin into the survival Forest Plot and have included both Hb and Albumin into the demographics table.
- Assessment of frailty and sarcopenia often includes grip tests or other assessments. Were any of these used beyond GO2 Frailty Grouping?
Unfortunately grip strength testing and other assessments were not included. The only other measure of note is the Get Up and Go test which was performed at baseline and a positive result was recorded if time taken was greater than 10 seconds or if the patient was unable to complete the test. Within the study population, 123 (51%) patients ‘failed’ the test. There was no difference in cBNP between those passing and those failing the get up and go test (p=0.88). There was also no significant different in OS (pass 7.62 months vs fail 6.6 months). We have therefore opted not to include this in the manuscript.
- Please list patients’ comorbidities, including cardiac.
No details regarding specific comorbidity was collected within the trial. However to be eligible, patients were required to have adequate renal function (GFR ≥30 ml/min (estimated or measured)) and hepatic function (bilirubin <3 times upper limit of normal (xULN). It could also be inferred that, as capecitabine was part of the regimen unlikely to have significant cardiovascular disease.
To try to address this lack of detail we have now included the following paragraph within the results:
‘Within the included population, 123 (51%) were unable to complete the timed up and go test within 10 seconds, 172 (71.4%) were taking five or more medications and 11 (4.6%) reported two or more falls in the previous 6 months.’
- Figure 3: Please explain how older patients have IMPROVED overall survival. I would expect worsened survival.
We believe the answer lies in disease biology as well as clinician bias. The younger patients recruited may have had more aggressive disease represented by being frail enough to be considered for the trial. This is what we observe anecdotally and we already know that in other tumour groups the biology of disease differs between older and younger patients. In addition, clinicians would have been happy to recruit older patients without frailty.
- Figure 3: The female sex is usually protective, but here it is worsening survival. Please recheck the data.
The data has been rechecked and it is a true representation. We agree – normally female sex appears to be protective and we have published on this previously (Baxter MA, et al. (2023) PMID: 37174057), however in this cohort it was not. This may be because of relatively small numbers included in this study.
- How do you explain that metastatic vs non-metastatic cancer did not experience any difference in survival?
Yes, no significant difference in overall survival was observed in the Forest plot but when looked at in isolation, there was inferior survival with metastatic disease (median 6.6 months vs 8.5 months, HR 0.66 (95% CI 0.47-0.91, p=0.013).
- What were the causes of mortality: cancer vs. comorbidity?
Patient were followed up for 12 months. Cause of death was not recorded, however in such an aggressive disease, death is most commonly due to progression of the cancer rather than comorbidity.
- Please shorten discussion
Thank you – we have tried to make the discussion more concise.
Reviewer 2 Report
Comments and Suggestions for Authors
The Authors find an important association between N-terminal pro-B-type Natriuretic Peptide in older adults with advanced gastroesophageal cancer. The manuscript is well written and sound and deserves publication.
Author Response
No comments – very positive
Dear Reviewer 2, thank you for taking the time to review our manuscript.
Reviewer 3 Report
Comments and Suggestions for Authors
This study sugests that the concentration of serum N-terminal pro-B-type Natriuretic Peptide (NT-proBNP) is associated with the outcome of the cytotoxic therapy for advanced/metastatic gastroesophageal cancer. To my opinion, the clarity of the manuscript is insufficient.
The study is relevant only to elderly/frail patients, however the Title does not reflect this fact.
Simple summary and Abstract should describe the rationale of the study. Why NT-proBNP was considered? What is the GO2 trial? Some of the messages given in the Abstract are misleading. For example, the GO2 trial included 559 subjects, however, in fact, only 241 were relevant to this paper. So, mentioning the former figure may result is some confusion There is an excessive number of unnecessary details in the Abstract, while essential details, like the number of patients in each NT-proBNP group (low, medium, high) is missing. “Further investigation in needed…” is not a sound conclusion… .
Please explain the hypothesis of the study in the Introduction.
Results
The Figure 1 compares “normal” vs. “high” levels of the NT-proBNP, while Figure 2 describes the comparison of “low” vs. “high” values. Please describe the rationale for performing these two analyses.
Why there was no correlation between the concentration of NT-proBNP and the performance status?
Lines 175-177: please provide numerical data, p values are not sufficient.
Author Response
Reviewer 3
This study sugests that the concentration of serum N-terminal pro-B-type Natriuretic Peptide (NT-proBNP) is associated with the outcome of the cytotoxic therapy for advanced/metastatic gastroesophageal cancer. To my opinion, the clarity of the manuscript is insufficient.
Dear Reviewer 3, thank you for taking the time to review our manuscript and provide feedback, which we feel has helped to significantly improve the content. Please find below our replies.
The study is relevant only to elderly/frail patients, however the Title does not reflect this fact.
We agree. We felt the previous title addressed this by mentioning ‘older adults’. We have now added frail to the title and hope that this improves clarity.
Simple summary and Abstract should describe the rationale of the study. Why NT-proBNP was considered? What is the GO2 trial? Some of the messages given in the Abstract are misleading. For example, the GO2 trial included 559 subjects, however, in fact, only 241 were relevant to this paper. So, mentioning the former figure may result is some confusion There is an excessive number of unnecessary details in the Abstract, while essential details, like the number of patients in each NT-proBNP group (low, medium, high) is missing. “Further investigation in needed…” is not a sound conclusion… .
Thank you for these comments. We have included the sentence in the simple summary ‘Serum NT-proBNP can be used as a measure of cardiac health and function and has established links with sarcopenia, which is prognostic in cancer’, which we hope explains the rationale. A similar sentence is included in the abstract.
We agree regarding the confusion of including the number 559 – we have therefore removed it.
We have tried to refine the abstract and add the pertinent details you noted.
We however disagree with your comments regarding the conclusion. We do not feel we can draw any firm conclusions from a post-hoc study and therefore we believe further investigation and validation of our results regarding the prognostic role of BNP is required.
Please explain the hypothesis of the study in the Introduction.
We have added the sentence ‘We hypothesized that an elevated baseline NT-proBNP would be associated with an inferior overall survival in those who received systemic chemotherapy’ to the end of introduction to specifically state our hypothesis.
Results
The Figure 1 compares “normal” vs. “high” levels of the NT-proBNP, while Figure 2 describes the comparison of “low” vs. “high” values. Please describe the rationale for performing these two analyses.
Thank you. Figure 1 addresses the initial question regarding whether an elevated BNP level is prognostic. When we determined this, we investigated whether level of elevation adds further prognostic value (Figure 2), which it does – i.e. those with the lowest level do better than those with a moderate level vs those with the highest. For visualisation, we have only included the low and high groups on Figure 2. To improve clarity, we have expanded a sentence in the results to explain this: ‘Further survival analysis to determine the impact of level of elevation demonstrated…’.
Why there was no correlation between the concentration of NT-proBNP and the performance status?
We were also surprised by this but we also recognise that ECOG PS is determined at a single timepoint, focuses on a single domain and is known to be a poor marker of overall frailty/deficit. It is also possible that ECOG PS could be impacted for a number of reasons both related to frailty and to cancer itself.
Lines 175-177: please provide numerical data, p values are not sufficient.
We have now provided these in an additional Table – Table 2.
Round 2
Reviewer 1 Report
Comments and Suggestions for Authors
My questions have been answered satisfactorily. Thank you
Reviewer 3 Report
Comments and Suggestions for Authors
-